# Extract of *Boehmeria nivea* Suppresses Mast Cell-Mediated Allergic Inflammation by Inhibiting Mitogen-Activated Protein Kinase and Nuclear Factor-κB

**DOI:** 10.3390/molecules25184178

**Published:** 2020-09-12

**Authors:** Ji-Ye Lim, Ji-Hyun Lee, Bo-Ri Lee, Mi Ae Kim, Young-Mi Lee, Dae-Ki Kim, Jin Kyeong Choi

**Affiliations:** 1Department of Immunology and Institute of Medical Sciences, Medical School, Jeonbuk (Chonbuk) National University, Jeonju, Jeonbuk 54907, Korea; 84juce@naver.com (J.-Y.L.); jihyunsh1211@naver.com (J.-H.L.); 2Department of Oriental Pharmacy, College of Pharmacy and Wonkwang-Oriental Medicines Research Institute, Wonkwang University, Iksan, Jeonbuk 54538, Korea; leebori1004@naver.com (B.-R.L.); ymlee@wku.ac.kr (Y.-M.L.); 3Department of Third Medicine, Professional Graduate School of Korean Medicine, Wonkwang University, Iksan, Jeonbuk 54538, Korea; cckma@hanmail.net

**Keywords:** *Boehmeria nivea*, allergic inflammation, cytokines, immunoglobulin E, mast cells

## Abstract

Mast cells are effector cells that initiate allergic inflammatory immune responses by inducing inflammatory mediators. *Boehmeria nivea* (Linn.) Gaudich is a natural herb in the nettle family Urticaceae that possesses numerous pharmacological properties. Despite the various pharmacological benefits of *Boehmeria nivea*, its effects on allergic inflammation have not yet been determined. Here, we investigated the effect of the ethanol extract of *Boehmeria nivea* (BNE) on degranulation rat basophilic leukemia (RBL)-2H3 mast cells stimulated with anti-dinitrophenyl (anti-DNP) and bovine serum albumin (BSA) during immunoglobulin E (IgE)-mediated allergic immune response. The results showed inhibition of the release of β-hexosaminidase and histamine from the cells. BNE suppressed pro-inflammatory cytokines (Tumor necrosis factor (TNF)-α, Interleukin (IL)-1β, and IL-6) and reduced T helper (Th)2 cytokine IL-4 expression and/or secretion correlated with the downregulation of p38, extracellular signal-regulated kinases (ERK) mitogen-activated protein kinase (MAPK), and nuclear factor-κB (NF-κB) signaling pathways in treated RBL-2H3 mast cells. In passive cutaneous anaphylaxis, treatment with BNE during IgE-mediated local allergic reaction triggered a reduction in mouse ear pigmentation and thickness. Taken together, these results indicated that BNE suppressed mast cell-mediated inflammation, suggesting that BNE might be a candidate for the treatment of various allergic disorders.

## 1. Introduction

Mast cells have been recognized to be involved in various types of allergic reactions including inflammatory disorders [1]. They are located at strategic locations, such as in the skin, mucosa, respiratory system, and gastrointestinal tract, and upon activation, mast cells express high-affinity Fc receptors (FcεRI) for binding specific immunoglobulin E (IgE) molecules on their surface [1]. Within IgE-dependent activation, degranulation leads to the release of histamine and generates other mediators stored in mast cell granules [2]. In addition, activated mast cells secrete inflammatory cytokines (Tumor necrosis factor (TNF)-α, Interleukin (IL)-1, IL-6, and T helper (Th)2) that contribute to the initiation of allergic reactions by recruiting and activating eosinophils, neutrophils, and Th2 cells, as well as interacting with tissue cells [3]. These mediators are associated with risk factor responses in allergic diseases that result from widespread mast cell activation, which may lead to the potentially life-threatening outcome of anaphylaxis [2,4]. Therefore, these mediators are the major target of effective therapies for allergic diseases. Although antihistamines and steroids are effective therapies for allergic diseases, serious adverse effects preclude their prolonged use. Therefore, an effective and a safe treatment remains a medical need.

*Boehmeria nivea* (Linn.) Gaudich is a plant in the nettle family Urticaceae, native to eastern Asia. This plant is used in traditional Chinese herbal medication and possesses various pharmacological properties [5]. The leaves are commonly used for treating wounds, while the roots are used in the treatment of colic in pregnancy, impetigo, threatened abortions, and leucorrhoea. The leaves in *Boehmeria nivea* contain numerous physiological active components, such as flavonoids, vitamins, minerals, rutin, and carotenoids, and have been found to exert a wide range of anti-inflammatory [6,7], anti-viral [5], anti-bacterial [8], anti-tumor [9], and anti-oxidant effects [10,11]. Despite the benefits of *Boehmeria nivea* and its potential use in various treatments, its underlying mechanism and effects on allergic reactions are unidentified. Therefore, the present study aimed to clarify the effects of the ethanol extract of *Boehmeria nivea* (BNE) on allergic inflammation and determine the mechanisms underlying these effects.

## 2. Results

### 2.1. BNE Inhibits Mast Cell Degranulation

Polyphenol compounds are the considered the best natural materials from an anti-inflammatory and anti-allergic perspective [12]. To study the potential regulatory role of BNE in allergic reactions and as a treatment for allergic disorders, we identified the phenolic content and major ingredients of BNE using HPLC chromatograms. The results of the HPLC profile of BNE indicated that rutin, luteorin-7-glucoside, naringin, and hesperidin were the major phenolic components of BNE (Figure 1A). This result was consistent with a previous study [6]. We next performed a 3-(4,5-Dimethylthiazol-2-yl)-2,5-Diphenyltetrazolium Bromide (MTT) assay to rule out the cell toxicity effects of BNE. Up to 200 μg/mL of BNE did not cause cell toxicity in RBL-2H3 cells after 24 h of exposure (Figure 1B). Mast cells are major sources of histamine and play a critical role in allergic reactions [13]. Inhibition of mast cell degranulation by certain agents is a therapeutic strategy for allergic diseases. We used rat basophilic leukemia (RBL)-2H3 (mast cell-like basophilic leukemia cells) to identify the effect of BNE on the degranulation of mast cells. Histamine release was rapidly induced by the dinitrophenyl-bovine serum albumin (DNP-BSA) challenge of anti-dinitrophenyl immunoglobulin E (anti-DNP IgE)-pre-activated RBL-2H3 cells, and BNE diminished histamine release (Figure 1C). BNE also inhibited β-hexosaminidase secretion, another component in the mast cell granule, in a dose-dependent manner (Figure 1D).

### 2.2. BNE Reduces Mast Cell-Mediated Inflammatory Cytokine Production 

Mast cell-mediated inflammatory cytokines (e.g., TNF, IL-1, IL-4, and IL-6) occur as part of the tissue remodeling related to atopic dermatitis and allergic asthma, and in many other scenarios characterized by chronic allergic inflammation [3]. We therefore determined whether BNE could suppress inflammatory cytokines in mast cells. The gene expression of cytokines (TNF-α, IL-1β, IL-6, and IL-4) in DNP-BSA-stimulated RBL-2H3 cells increased tremendously approximately 3.5- to 12-fold compared to unstimulated cells, and treatment with BNE reduced their expression in a dose-dependent manner (Figure 2A). We also confirmed the production of TNF-α, IL-1β, IL-6, and IL-4 in the supernatants of the activated RBL-2H3 cells by ELISA. Similar to the gene expression results, BNE suppressed production of these cytokines (Figure 2B).

### 2.3. BNE Suppresses Inflammatory Cytokines by Downregulating p38, ERK MAPK, and Nuclear Factor-κB (NF-κB) Pathways 

FcεRI ligation induces mitogen-activated protein kinase (MAPK) (p38, extracellular signal-regulated kinases; ERK, and c-Jun N-terminal kinase; JNK) and NF-κB pathway activation, which can generate inflammatory cytokines such as TNF-α, IL-1, IL-4, and IL-6 in mast cells [14]. Therefore, we investigated whether BNE might have inhibited mast cell-mediated allergic reaction by inhibiting MAPKs and NF-κB downstream from FcεRI aggregation. RBL-2H3 cells were pretreated for 1 h with BNE. After 1 h, the cells were stimulated for an additional hour in a culture medium containing DNP-BSA. Western blot analysis revealed that BNE suppressed the activation of the p38 and ERK pathway; however, BNE could not downregulate the JNK pathway (Figure 3A). Moreover, the nuclear translocation of NF-κB and degradation of I kappa-B alpha (IκBa) in DNP-BSA-stimulated RBL-2H3 cells were significantly inhibited by BNE (Figure 3B), suggesting that BNE suppresses mast cell-mediated inflammatory cytokines (Figure 2) by inhibiting p38, ERK, and NF-κB pathways. 

### 2.4. BNE Attenuates IgE-Mediated Local Cutaneous Anaphylaxis Reaction 

We used passive cutaneous anaphylaxis, a type I hypersensitivity in vivo model, and examined whether treatment with BNE would ameliorate allergic reactions. One day after anti-DNP IgE sensitization, a DNP-BSA-4% Evans blue mixture was intravenously injected into each mouse to induce a systemic allergic response. Consequently, the vascular permeability expanded and allowed for leakage of the Evans blue dye to produce a blue spot at the ear intradermal injection site. Mice that were injected with anti-DNP IgE antigen from phosphate buffered saline (PBS)-treated mice showed increased pigmentation due to relatively high vascular permeability, while mice that were orally administered BNE (100 or 200 mg/kg) and dexamethasone (positive control; 10 mg/kg) had reduced ear pigmentation (Figure 4A). Next, the pigmented dye was extracted from each mouse ear and quantified by measuring the absorbance. The BNE treatment resulted in significantly lower absorbance values than those observed in PBS-administered mice, similar to the dexamethasone group (Figure 4B). Oral administration of BNE also decreased ear thickness (Figure 4C).

## 3. Discussion

Phytochemical studies on *Boehmeria nivea* leaves have reported that they contain polyphenolic compounds such as caffeic acid, catechin, epicatechin, β-sissterol, rutin, luteolin-7-glucoside, naringin, hesperidin, chlorogenic acid, and tangeretin [6,15,16]. These are known to relieve allergic diseases such as atopic dermatitis, asthma, and rhinitis [17,18,19]. Consistent with previous studies, we found that *Boehmeria nivea* leaves contained abundant polyphenol components. Recent studies have shown that BNE exerts anti-inflammatory [6], antioxidant [15], anti-hepatitis B virus [5], anti-colitis [7], anti-obesity [20], and anti-diabetic effects [21]. Based on these known effects, we assumed that BNE might play a role in suppressing IgE-mediated mast cell inflammation and could be used for treating allergic diseases in the future.

IgE- and mast cell-dependent reactions can lead to chronic allergic diseases, with tissue damage and remodeling [22]. In this study, we have shown that BNE possesses inflammatory mediator-suppressive activities during IgE-engaged mast cell activation and can therefore be a potential treatment for allergic diseases. Mast cells release various mediators after cross-linking of the IgE receptors on their surface and can induce an immediate allergic reaction. The procedure of degranulation in mast cells comprises fusing of the membrane of the granules, including biogenic amines such as histamine, lysosomal enzymes such as β-hexosaminidase, lipid mediators, and cytokines [23]. Experimentally, mast cell degranulation can be activated via FcεRI by using anti-IgE [24]. In allergic reactions, histamine causes vasodilation and capillary leakage and induces fever. Antihistamine is mainly used in the treatment of allergic diseases; however, in some allergic disorders, antihistamines are not effective at inhibiting the response [22]. As indicated in Figure 1, BNE suppressed histamine degranulation and β-hexosaminidase secretion. Thus, treatment with BNE can be developed as a candidate for allergic disease therapy, with similar effects as antihistamines but no side effects.

Mast cells produce many different cytokines that contribute to the late-phase of allergic inflammation, and production of cytokines by activated mast cells is an important consequence of newly induced gene transcription [22]. Recruitment and activation of various adaptor molecules and kinases in response to FcεRI cross-linking lead to the activation of both NF-κB [22] and MAPK pathways (e.g., p38, ERK, and JNK) [25,26]. These transcription factors stimulate mast cell degranulation and the expression of several cytokines such as TNF, IL-1, IL-4, and IL-6 [22]. The cytokine gene promoter involves NF-κB, elements that control gene transcription [27], and targeting by the MAPK pathways containing p38 and JNK [25,26]. BNE reduces the production of inflammatory cytokines by downregulating the p38 and JNK MAPK pathways in macrophages [6]. However, our present data revealed that BNE downregulated p38 and ERK phosphorylation but not the activation of JNK MAPK. The p38 and JNK MAPK pathways are often concurrently activated, but it appears to depend on the extracellular stimulus and cell type [25,28]. Thus, BNE can suppress the production of inflammatory cytokines by selectively reducing the activation of the p38 MAPK pathway in mast cells. These results can be an extension of further research: BNE can challenge the role of histone modification to the regulation of the pathogenesis of allergic diseases with an explanation of the molecular mechanisms [29,30]. For example, BNE can contribute to various allergic disorders by inhibiting the induction of H3S10ph, similar to the p38 MAPK inhibitor [30,31]. In contrast, our data can support the potential of BNE treatment to suppress proinflammatory cytokine expression and the synthesis of other lipid mediators through the downregulation of the ERK MAPK pathway in mast cell-mediated allergic diseases. Numerous studies have revealed that NF-κB is a crucial transcription factor in allergic disorders, such as allergies, asthma, and atopic dermatitis [32,33,34]. NF-κB activation does appear to be vital for the optimal production of certain mast cell-derived cytokines, especially IL-6 [35]. IL-6 induces histamine production in mast cells and is an important inducing factor for the expression of FcεRI [36]. In the present study, BNE reduced NF-κB activation and the production of IL-6 and histamine. Thus, treatment with BNE could downregulate the NF-κB pathway and in turn diminish IL-6 expression to reduce histamine production. 

Passive cutaneous anaphylaxis is a well-established experimental animal model for demonstrating type I hypersensitivity responses [37]. The early vascular changes that happen during immediate hypersensitivity responses are indicated by the wheal-and-flare response to the intradermal injection of an IgE allergen in the mouse ear [22]. In response to IgE-sensitized release of mast cell mediators, local blood vessels first dilate and then leak fluid and macromolecules, which results in redness and local swelling, followed by dermal activation of mast cells with a release of inflammatory mediators, notably histamine [22]. Our data showed that treatment with BNE improved the ear pigmentation and swelling caused by allergens, indicating that BNE controlled the activation of mast cells in the skin. 

In this study, the HPLC profile of BNE indicated that rutin, luteolin-7-glucoside, naringin, and hesperidin are the major ingredients of BNE (Figure 1A). Among the main components of BNE, rutin inhibited histamine release, decreased proinflammatory cytokines, and suppressed activation of NF-κB in mast cells [38]. We also demonstrated the effect of rutin on allergic inflammatory diseases [18]. However, luteorin-7-glucoside, naringin, or hesperidin did not significantly affect in mast cell-mediated inflammation [39,40]. Therefore, the effect of BNE on mast cell inflammation, we speculate, was mostly contributed by rutin.

Currently, therapies to control mast cell activation in patients or offset mast cell effects have not yet been completely developed [2]. To date, BNE has been used in East Asia as an herbal medicine for various treatments and is considered relatively safe; the present findings and those of many previous studies have not shown any toxicity. Moreover, we provide evidence that BNE could be a potential therapeutic candidate for controlling mast cell activation. It is vital to develop a safe and well-designed treatment for mast cell-mediated allergic reactions.

## 4. Materials and Methods

### 4.1. Reagents

Anti-DNP IgE, DNP-bovine serum albumin (BSA), dexamethasone, and Evans blue were purchased from Sigma (St. Louis, MO, USA). The RBL-2H3 cell line (KCLB-22256) was purchased from the Korean Cell Line Bank (Seoul, Korea). Dulbecco’s modified Eagle’s medium (DMEM), fetal bovine serum, and penicillin-streptomycin (PS) were purchased from Hyclone (Logan, UT, USA). The histamine, TNF-α, IL-6, IL-1β, and IL-4 ELISA kits were purchased from Biovision (Milpitas, CA, USA). Anti-phospho-p38 (Thr180/Tyr182), anti-phospho-ERK (L352), anti-phospho-JNK (T183/Y185), anti-p38, anti-ERK, anti-JNK, and β-actin were purchased from Santa Cruz Biotechnology (Santa Cruz, CA, USA). *Boehmeria nivea* is an eco-friendly pesticide-free product, which was purchased from a farming association (Yeonggwang, Korea).

### 4.2. Preparing BNE and HPLC Analysis

*Boehmeria nivea* leaves were ground in a blender, and the powders were extracted with 70% ethanol. After precipitation, the supernatants were lyophilized using a freeze dryer (Ilshin, Dongduchun, Korea) to remove residual ethanol. After adding 5 mL of 50% methanol to the prepared 100 mg of *Boehmeria nivea* powder, ultrasonic shaking was performed for 30 min, followed by centrifugation at 3300 rpm for 15 min. The supernatant was removed and re-extraction was performed, and the supernatant obtained after two extractions was combined to make up a volume of 10 mL. Chromatographic analysis was performed using an Aglient 1260 infinity II (Santa Clara, CA, USA) high performance liquid chromatography (HPLC) system with a Waters 996 photodiode detector. For quantification based on the internal standard method, a UV detector at 260 nm was used with a ZORBOX Eclipse Plus C18 (4.6 mm × 250 mm, 5.0 μm) column (Agilent, Santa Clara, CA. USA). The eluent was gradient of 0.1% acetic acid in water (A) and 0.1% acetic acid in acetonitrile (B) at a flow rate of 1 mL/min. The gradient was as follows: 0 min, 88:12% A:B; 18 min, 78:22% A:B; 21 min, 74:26%, and A:B; 25 min, 72:28%. The analysis results are shown in Figure 1. 

### 4.3. Animals

Six-week-old male BALB/c mice were purchased from SAMTAKO Bio Korea. Before the experiment, all mice were allowed to adapt to the environment for 7 days. Drinking water was provided to the mice ad libitum, and the mice were divided into five groups (*n* = 6). During the study, the mice were housed in a room at a temperature of 22 ± 2 °C and humidity of 55 ± 5%. The management and treatment of mice were conducted under the approval of the Institutional Animal Care and Use Committee, Jeonbuk National University (JBNU-2020-087).

### 4.4. IgE-Mediated Passive Cutaneous Anaphylaxis

The IgE-mediated passive cutaneous anaphylaxis model was established as described previously [23]. The ear skin of the mouse (*n* = 6/group) was sensitized by intradermal injection of anti-DNP-IgE (0.5 µg/site) using an insulin syringe. After 48 h, BNE (100 mg/kg and 200 mg/kg) and dexamethasone (10 mg/kg) were injected intraperitoneally, except in the normal group. DNP-BSA (10 mg/mL) was mixed with 4% Evans blue (1:1) and injected into the eye vein. Thirty minutes after the challenge, the thickness of the ears was measured, and mice were anesthetized by CO_2_. The ear tissues of the stained area were cut and immersed in 1 mL KOH (1 M) and incubated for 2 days to melt. Then, 4 mL of acetone-phosphoric acid mixture (5:13) was added to the incubated tube to stop the reaction. After vortexing and centrifugation, the absorbance was measured at 620 nm using a spectrophotometer.

### 4.5. MTT Assay

The RBL-2H3 cells were seeded onto a 96-well plate at a density of 1 × 10^4^ cells/well and incubated in DMEM containing FBS and PS for 24 h. Cells were treated with different concentrations of BNE (25 μg/mL to 250 μg/mL) for 24 h. MTT analysis was performed to confirm the cell viability of RBL-2H3 cells. Formazan crystallized by treating with 50 µg/mL MTT solution was dissolved in DMSO and measured at 540 nm using a microplate reader.

### 4.6. β-Hexosaminidase and Histamine Release

To assess the anti-allergic effects of BNE, two indicators of degranulation, β-hexosaminidase and histamine release, were investigated. RBL-2H3 cells (6 × 10^5^ cells/mL) were seeded onto 48-well plates and sensitized with 100 ng/mL anti-DNP-IgE for 24 h. The anti-DNP IgE-sensitized RBL-2H3 cells were pre-incubated for 1 h with BNE (25 μg/mL to 250 μg/mL) and stimulated by DNP-BSA (100 ng/mL) for 4 h. The culture supernatant was collected, the protein was precipitated, and the supernatant was mixed with the substrate solution and then incubated at 37 °C for 1.5 h. After the reaction was terminated by adding a stop solution (0.1 M Na_2_CO_3_/NaHCO_3_, pH 10.0) to each well, the absorbance was confirmed at 405 nm using a microplate reader (BioTek, VT, USA). The histamine concentration was measured using an ELISA kit according to the manufacturer’s instructions after collecting the supernatant of the culture following stimulation with DNP-BSA for 6 h under the same conditions.

### 4.7. Inflammatory Cytokine Assay

To measure the release of pro-inflammatory cytokines, a culture supernatant of RBL-2H3 cells stimulated with DNP-BSA was collected and subjected to ELISA. The levels of TNF-α, IL-1β, IL-6, and IL-4 were assessed according to the manufacturer’s instructions.

### 4.8. Real-Time Polymerase Chain Reaction

Cell culture and RNA isolation were performed as described previously [23]. The RBL-2H3 cells were stimulated with DNP-BSA for 1 h and harvested. After RNA extraction, cDNA synthesis was performed using a PrimeScript™ II 1st Strand cDNA synthesis kit. Quantitative amplification was performed using the AB StepOne system (Applied Biosystems, Foster, CA, USA). The following primer sequences were used: *rTnfa* (F5′-GAAAGCATGATCCGAGATGTGG-3′, R5′-TCATACCAGGGCTTGAGCTCA-3′), *rIl1β* (F5′-CCCTGCAGCTGGAGAGTGTGG-3′, R5′-TGTGCTCTGCTTGAGTGCT-3′), *rIl6* (F5′-GGAGACTTCACAGAGGATAC-3′, R5′-CCATTAGGAGAGCATTGGAAG-3′), *rIl4* (F5′-ACCCTGTTCTGCTTTCTC-3′, R5′-GTTCTCCGTGGTGTTCCT-3′), and *rGapdh* (F5′-AACGGCACAGTCAAGGCTGA-3′, R5′-ACGCCAGTAGACTCCACGACAT-3′).

### 4.9. Western Blot Analysis

Western blotting was performed as described previously [23]. Nuclear and Cytoplasmic samples of cells were performed using a nuclear and cytoplasmic extraction reagents kit (Thermo fisher, Waltham, MA, USA) according to the manufacturer’s protocols. Briefly, the same amount of protein was separated using 10% SDS page, electrophoresed at 110 V for 90 min, and transferred to a polyvinylidene fluoride (PVDF) membrane. Protein phosphorylation or expression was detected using specific antibodies, and this was visualized using chemiluminescence reagents.

### 4.10. Statistical Analysis

Statistical analysis was performed using GraphPad Prism 8 (San Diego, CA, USA). After using the unidirectional ANOVA, the treatment effect was analyzed using Tukey’s multiple comparison test. All data are expressed as the mean ± standard error of the mean (SEM). *p* < 0.05 values were considered statistically significant.

## 5. Conclusions

We found that BNE treatments were critical regulators of allergic reactions and could suppress histamine and β-hexosaminidase production in mast cells. However, the effects of BNE on various allergic diseases remain unknown. Moreover, the suppressive effect of BNE on different allergic disease models, such as atopic dermatitis and asthma, should be demonstrated. Nonetheless, the discovery that BNE can inhibit inflammatory cytokines by downregulating MAPK and NF-κB pathways indicates the therapeutic potential of BNE and could facilitate investigations into its effects on various allergic diseases.

## Figures and Tables

**Figure 1 molecules-25-04178-f001:**
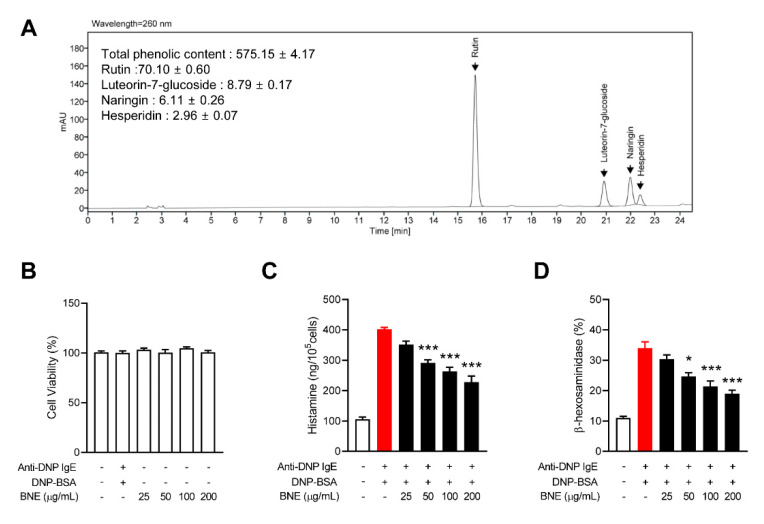
Effects of BNE on the degranulation of mast cells. (**A**) HPLC analysis of phenolic compounds. The peaks shown in the HPLC chromatogram results: rutin, luteorin-7-glucoside, naringin, and hesperidin. (**B**) Rat basophilic leukemia (RBL)-2H3 cells were treated with indicated concentrations of BNE for 12 h. The cell viability was measured using the MTT assay. (**C**,**D**) RBL-2H3 (6 × 10^5^ cells/well) cells were pre-sensitized with anti-DNP IgE (100 ng/mL) overnight. The cells were treated with or without indicated doses of BNE for 1 h and then challenged with anti-DNP-BSA for 4 h. The histamine (**C**) and β-hexosaminidase (**D**) levels in cultured media were measured as described in the Materials and Methods section. Data represent the mean ± standard error of the mean (SEM) from three independent experiments. * *p* < 0.05; *** *p* < 0.001, Tukey’s test, significantly different from Anti-DNP IgE plus DNP-BSA-treated group. Anti-DNP IgE, anti-dinitrophenyl immunoglobulin E; DNP-BSA, dinitrophenyl-Bovine serum albumin; BNE, *Boehmeria nivea* extract.

**Figure 2 molecules-25-04178-f002:**
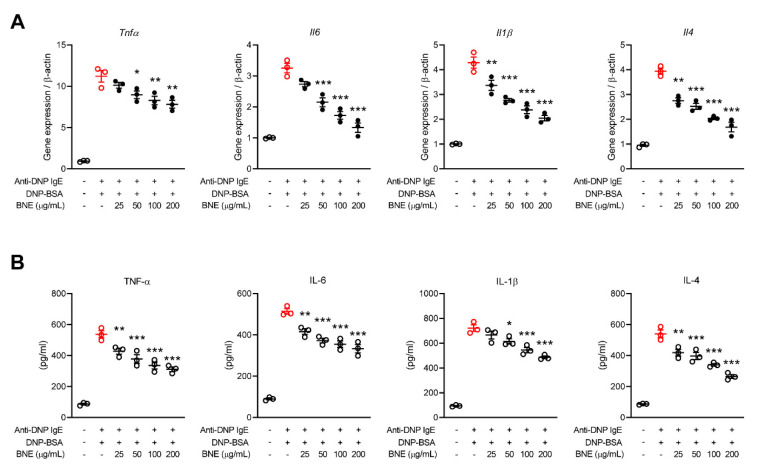
Effects of BNE on the expression and/or secretion of inflammatory cytokines. (**A**,**B**) RBL-2H3 cells were pre-sensitized with anti-DNP IgE (5 × 10^5^ cells/well). The cells were activated with anti-DNP-BSA by pretreating with or without DNE, and qPCR was used to quantify the expression of inflammatory cytokine (TNF-α, IL-1β, IL-6, and IL-4) genes (**A**). (**B**) Analysis of cytokines in the cultured supernatant for 6 h by ELISA. Data represent the mean ± SEM from three independent experiments. * *p* < 0.05; ** *p* < 0.01; *** *p* < 0.001, Tukey’s test, significantly different from Anti-DNP IgE plus DNP-BSA-treated group. Anti-DNP IgE, anti-dinitrophenyl immunoglobulin E; DNP-BSA, dinitrophenyl-Bovine serum albumin; BNE, *Boehmeria nivea* extract; TNF-α, tumor necrosis factor-alpha; IL, interleukin.

**Figure 3 molecules-25-04178-f003:**
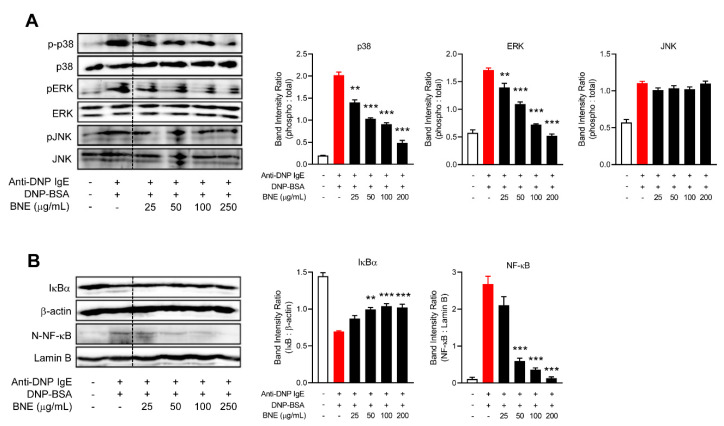
BNE inhibited MAPKs and the nuclear factor-κB (NF-κB) signaling pathway. (**A**,**B**) RBL-2H3 cells were pre-sensitized with anti-DNP IgE (1 × 10^6^ cells/well). Cells were pre-sensitized with anti-DNP IgE (100 ng/mL) overnight. The cells were treated with or without indicated doses of DNE for 1 h and then challenged with anti-DNP-BSA for 15 min in RBL-2H3. Whole cell lysates or nuclear extracts were analyzed by western blotting: MAPKs (p38, ERK, and JNK) (**A**) and NF-κB (**B**) pathways. Protein levels were normalized to total form or β-actin and quantified using Image-J software (right panel). Data represent the means ± SEM from three independent experiments. ** *p* < 0.01; *** *p* < 0.001, Tukey’s test, different from the Anti-DNP IgE plus DNP-BSA-treated group. Anti-DNP IgE, anti-dinitrophenyl immunoglobulin E; DNP-BSA, dinitrophenyl-Bovine serum albumin; BNE, *Boehmeria nivea* extract; MAPKs, mitogen-activated protein kinase; NF-κB, nuclear factor-κB; IκB, I kappa-B alpha.

**Figure 4 molecules-25-04178-f004:**
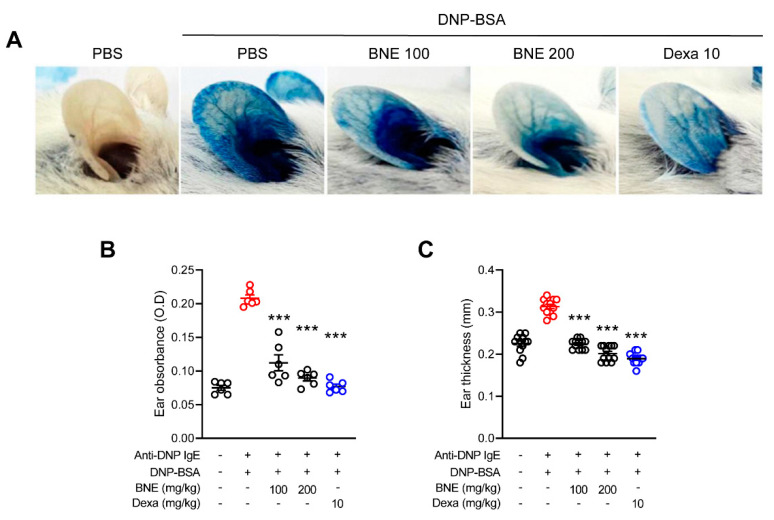
Effects of DNE on IgE-mediated passive cutaneous anaphylaxis. (**A**) The ear skin of mice (*n* = 6/group) was sensitized by intradermal injection of anti-DNP IgE (0.5 mg/site) for 48 h. DNE (100 and 200 mg/kg) or dexamethasone (10 mg/kg) was orally administered before the intravenous (i.v.) injection of a DNP-BSA and 4% Evans blue (1:1) mixture. After 30 min, the thickness of both ears was measured. (**B**) Ear thickness was measured with a dial thickness gauge. (**C**) The dye was extracted and quantified using a spectrophotometer. Data represent the mean ± SEM from three independent experiments. *** *p* < 0.001, Tukey’s test, significantly different from Anti-DNP IgE plus DNP-BSA with PBS-treated group. Anti-DNP IgE, anti-dinitrophenyl immunoglobulin E; DNP-BSA, dinitrophenyl-Bovine serum albumin; BNE, *Boehmeria nivea* extract; Dexa, dexamethasone.

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
