# Peer review of "Extract of Boehmeria nivea Suppresses Mast Cell-Mediated Allergic Inflammation by Inhibiting Mitogen-Activated Protein Kinase and Nuclear Factor-κB"

_molecules, 2020, doi:10.3390/molecules25184178_

Round 1
Reviewer 1 Report
The manuscript written by Lim et al. shows that an aqueous methanolic extract from Boehmeria nivea (BNE) suppresses degranulation and expression of proinflammatory cytokines in rat basophilic leukemia cell line RBL-2H3 cells. In addition, a single intraperitoneal injection of BNE attenuates the passive cutaneous anaphylaxis reaction in mice. Although the results looks interesting, the bioactive ingredient in BNE is not identified in this study. An effect of an extract is sometimes unreproducible because of several reasons. Fortunately, four major ingredients in BNE were identified and quantified in this study, which are commercially available. The authors thus need to evaluate the effect of each of and the mixture of the four compounds at least to identify the bioactive ingredient in BNE.
The authors stated in the Materials and Methods section that the effect was analyzed by Tukey's test; however, they mentioned in the figure legends of Figures 1 to 4 that the data were analyzed by Student's t-test. Because all data should be analyzed by a multiple comparison test, the authors should analyze all data by Tukey's test. In addition, the authors need to state in the figure legends of Figures 1 to 4 that which group was compared with asterisked groups.
The authors need to state that which part of the plant of Boehmeria nivea was used for preparing the extract. The authors also need to state that what kind of solvent was used in HPLC analysis. In addition, the authors need to describe the HPLC condition (gradient or isocratic) and the C18 column's name and company.
The authors need to describe how they prepared cytoplasmic and nuclear protein samples for Western blotting. In addition, the authors need to specify the phosphorylated amino acid residue of p38, ERK, and JNK in the Western blot analysis.
Author Response
Response to Reviewer 1 Comments
Point 1: The manuscript written by Lim et al. shows that an aqueous methanolic extract from Boehmeria nivea (BNE) suppresses degranulation and expression of proinflammatory cytokines in rat basophilic leukemia cell line RBL-2H3 cells. In addition, a single intraperitoneal injection of BNE attenuates the passive cutaneous anaphylaxis reaction in mice. Although the results looks interesting, the bioactive ingredient in BNE is not identified in this study. An effect of an extract is sometimes unreproducible because of several reasons. Fortunately, four major ingredients in BNE were identified and quantified in this study, which are commercially available. The authors thus need to evaluate the effect of each of and the mixture of the four compounds at least to identify the bioactive ingredient in BNE.
Response 1: We thank for Reviewer’s critical comments. We agree that among the four major components of BNE (rutin, luteolin-7-glucoside, naringin, and hesperidin), we should identify the main bioactive ingredients, as suggested by the reviewer. However, the four main components of BNE have already been demonstrated in previous studies. Among the main ingredients of BNE, only rutin showed that excellent effects in mast cell-mediated inflammation, such as proinflammatory cytokine suppression and histamine release (Arch Pharm Res. 2008;31:1303-11). We demonstrated the effect of rutin on allergic inflammatory diseases (Exp Bio Med. 2013, 238:410-7). Unfortunately, luteolin-7-glucoside, naringin, and hesperidin have not shown an effect in mast cell-mediated inflammation in vitro and in vivo (Chemistry of Natural Compounds. 2016; 52; J Nat Med. 2013; 67:643-646). Moreover, rutin content is approximately 10 times higher than that of the rest components among the four main ingredients in our results. Therefore, we believe that although BNE has four main components, most of the effects of BNE are from rutin. If the reviewer strongly requests the mixtures of the four compounds test, we may do it, however even if it is used by mixing each ingredient, we speculate to be the same results in our study. The results of previous studies on the efficacy of each of these ingredients were discussed in the Discussion section (line 218-224).
Point 2: The authors stated in the Materials and Methods section that the effect was analyzed by Tukey's test; however, they mentioned in the figure legends of Figures 1 to 4 that the data were analyzed by Student's t-test. Because all data should be analyzed by a multiple comparison test, the authors should analyze all data by Tukey's test. In addition, the authors need to state in the figure legends of Figures 1 to 4 that which group was compared with asterisked groups.
Response 2: We thank the Reviewer for pointing out our mistake during manuscript preparation. We corrected the statistical method. For mean scores, the Tukey’s test was used to identify significant differences between experimental groups compared with the anti-DNP IgE plus DNP-BSA. The data are presented as the mean ± SEM. This information has been added to the revised manuscript Legends for Figures.
Point 3: The authors need to state that which part of the plant of Boehmeria nivea was used for preparing the extract. The authors also need to state that what kind of solvent was used in HPLC analysis. In addition, the authors need to describe the HPLC condition (gradient or isocratic) and the C18 column's name and company.
Response 3: Following the reviewer’s suggestion, detailed Boehmeria nivea plant part and HPLC information has been added as a Materials and Methods section (line 243-255).
Point 4: The authors need to describe how they prepared cytoplasmic and nuclear protein samples for Western blotting. In addition, the authors need to specify the phosphorylated amino acid residue of p38, ERK, and JNK in the Western blot analysis.
Response 4: In response to reviewer’s suggestion, this information has been added in Materials and Methods section (line 238-239, 308-310).
Reviewer 2 Report
With interest, I read the manuscript molecules-926781.
Comments:
- Overall, the idea behind this manuscript is fine. The Authors start with identification of BNE components, then test them in vitro in cells, and then in vivo in animals.
- However, the Authors stay at a certain level of generalization. They characterize the composition of BNE and identify molecular effects of BNE in mast cells or “clinical” effects related to its application in mice but they do not make it clear which component of BNE is responsible for the effects or which component is responsible for which effect. Some speculations in the Discussion would be welcome.
- Likewise, some mechanistic speculations would be welcome to clarify how BNE components lead to altered mast cell secretional function. For example, the Authors could speculate on the potential involvement of epigenetic mechanisms, especially histone modifications, known to mediate multiple processes involved in the development of allergies (PMID: 29796022, 28322581).
- Sometimes, the Authors introduce abbreviations without explaining them, e.g. in line 43. Please, correct. The same applies to the figures. Newly introduced abbreviations should be separately explained in the abstract, main text, and the figures (e.g. “anti-DNP-BSA”).
- Figures 1-4. P-values are given for which comparisons. It should be made precise.
- Besides, description of the statistical methodology (lines 287-291) does not match the content of the manuscript, incl. the figures. ANOVA? Tukey’s test?
- In your experiments, you use the ethanol extract of Boehmeria nivea (BNE), i.e. Boehmeria nivea dissolved in ethanol, in different concentrations. I guess, to obtain those concentrations, you diluted in PBS or something like that? Thus, in your experiments with different BNE concentrations, was not only Boehmeria nivea concentration but also ethanol concentration different between the conditions?
- What was the effect of BNE on FcɛRI surface levels? Do you have such data?
Other comments:
- Lines 79 and 84. What is “DNE”?
- Line 190. “mast cell-mediated cytokines”?
Author Response
Response to Reviewer 2 Comments
Point 1: Overall, the idea behind this manuscript is fine. The Authors start with identification of BNE components, then test them in vitro in cells, and then in vivo in animals. However, the Authors stay at a certain level of generalization. They characterize the composition of BNE and identify molecular effects of BNE in mast cells or “clinical” effects related to its application in mice but they do not make it clear which component of BNE is responsible for the effects or which component is responsible for which effect. Some speculations in the Discussion would be welcome.
Response 1: Following the reviewer’s suggestion, the efficacy of each of these ingredients were discussed in the Discussion section (line 218-224).
Point 2: Likewise, some mechanistic speculations would be welcome to clarify how BNE components lead to altered mast cell secretional function. For example, the Authors could speculate on the potential involvement of epigenetic mechanisms, especially histone modifications, known to mediate multiple processes involved in the development of allergies (PMID: 29796022, 28322581).
Response 2: Thank you for this insight. In our revised manuscript, we referenced some of the above articles. If provided with another chance, we will incorporate the epigenetic mechanisms in a future study (line 196-199).
Point 3: Sometimes, the Authors introduce abbreviations without explaining them, e.g. in line 43. Please, correct. The same applies to the figures. Newly introduced abbreviations should be separately explained in the abstract, main text, and the figures (e.g. “anti-DNP-BSA”).
Response 3: To follow the reviewer's suggestion, we added the abbreviations in each Legends for Figures in the revised manuscript.
Point 4: Figures 1-4. P-values are given for which comparisons. It should be made precise.
Response 4: Following the reviewer’s suggestion, the comparison information has been added in the revised manuscript Legends for Figures.
Point 5: Besides, description of the statistical methodology (lines 287-291) does not match the content of the manuscript, incl. the figures. ANOVA? Tukey’s test?
Response 5: We are sorry for the reviewer’s confusion. We corrected the statistical method. For mean scores, the Tukey’s test was used to identify significant differences between experimental groups compared with the anti-DNP IgE plus DNP-BSA. The data are presented as the mean ± SEM. This information has been added to the revised manuscript Legends for Figures.
Point 6: In your experiments, you use the ethanol extract of Boehmeria nivea (BNE), i.e. Boehmeria nivea dissolved in ethanol, in different concentrations. I guess, to obtain those concentrations, you diluted in PBS or something like that? Thus, in your experiments with different BNE concentrations, was not only Boehmeria nivea concentration but also ethanol concentration different between the conditions?
Response 6: We used ethanol extraction because BNE does not melt well in water and melts in organic solvents. After extraction, the residual ethanol was dried entirely and removed, and the dried extract was dissolved in phosphate-buffered saline (PBS) according to the concentration to be used in vitro and in vivo experiments. Therefore, unlike the reviewer’s concerns, BNE is NOT being related to ethanol concentration in our experiment results. BNE extract information has been added in the Material and Methods section (line 243-245).
Point 7: What was the effect of BNE on FcɛRI surface levels? Do you have such data?
Response 7: Thank you for reviewer’s critical question. Allergen binding to IgE molecules occupying FcϵRI on the surface of RBL-2H3 cells results in cross-linking of these receptors and subsequent cell degranulation and mediator release, leading to the development of allergic symptoms typical for type I hypersensitivity reactions (J Allergy Clin Immunol. 2009;124:639-46). In fact, we did not check FceRI surface level data because we thought BNE adjusted subsequent inflammatory responses from mast cells by reducing FceRI. For this reason, numerous studies actually exclud direct FceRI surface results. However, it would be interesting to check the level of FceRI by extraction and purification of the main components in future studies. The reviewer's various insights will help our research in the future.
Other comments:
Point 1. Lines 79 and 84. What is “DNE”?
Response 1. The requisite corrections have been made in the revised manuscript.
Point 2. Line 190. “mast cell-mediated cytokines”?
Response 2. In response to reviewer’s, we changed the "mast cell-mediated" to "Mast cell-derived cytokines".
Round 2
Reviewer 1 Report
This reviewer considers that the authors addressed all concerns and improved their manuscript.
Reviewer 2 Report
My comments have been addressed well. Thank you.